# Electro-Exfoliation of Graphite to Graphene in an Aqueous Solution of Inorganic Salt and the Stabilization of Its Sponge Structure with Poly(Furfuryl Alcohol)

**DOI:** 10.3390/nano9070971

**Published:** 2019-07-03

**Authors:** Anna Ilnicka, Malgorzata Skorupska, Piotr Kamedulski, Jerzy P. Lukaszewicz

**Affiliations:** 1Faculty of Chemistry, Nicolaus Copernicus University, Gagarina 7, 87-100 Torun, Poland; 2Centre for Modern Interdisciplinary Technologies, Nicolaus Copernicus University, Wilenska 4, 87-100 Torun, Poland

**Keywords:** graphene, electrochemical exfoliation, aqueous solution, poly(furfuryl alcohol), Raman spectroscopy, graphite electrode

## Abstract

We demonstrate an accessible and effective technique for exfoliating graphite foil and graphite powder into graphene in a water solution of inorganic salt. In our research, we report an electrochemical cathodic exfoliation in an aqueous solution of Na_2_SO_4_. After electro-exfoliation, the resulting graphene was premixed with furfuryl alcohol (FA) and an inorganic template (CaCO_3_ and Na_2_CO_3_). Once FA was polymerized to poly(furfuryl alcohol) (PFA), the mixture was carbonized. Carbon bridges originating in thermally-decomposed PFA joined exfoliated graphene flakes and stabilized the whole sponge-type structure after the nano-template was removed. Gases evolved at the graphite electrode (cathode) played an important role in the process of graphene-flake splitting and accelerated the change of graphite into graphene flakes. Starting graphite materials and graphene sponges were characterized using Raman spectroscopy, SEM, high-resolution transmission electron microscopy (HRTEM), elemental analysis, and low-temperature adsorption of nitrogen to determine their structure, morphology, and chemical composition. The discovered manufacturing protocol had a positive influence on the specific surface area and porosity of the sponges. The SEM and HRTEM studies confirmed a high separation degree of graphite and different agglomeration pathways. Raman spectra were analyzed with particular focus on the intensities of I_D_ and I_G_ peaks; the graphene-type nature of the sponges was confirmed.

## 1. Introduction

Graphene is a two-dimensional layer of carbon atoms which form hexagonal rings based on sp^2^ hybridization [1,2]. A particularly promising graphene production technique is the creation of a colloidal suspension of graphene flakes or its derivatives [3]. Unlike other methods, e.g., chemical vapor deposition, epitaxial growth, and microchemical exfoliation, this approach is versatile in terms of chemical functionalization and affording the possibility of high-volume production. The latter is sometimes used to justify focus on obtaining graphene as a water dispersion [4]. Generally, exfoliation of graphene/graphene oxide is widely exploited and was a key step in the manufacturing of complex materials like bionanocomposite based on polylactic acid [5], graphene oxide and biodegradable polymer blends [6], graphene oxide (GO) and polyamide composites, and nanosilica modified graphene [7]. Recently, the electrochemical approach has been found to have the advantages of being single step, easy to operate, environmentally friendly, and able to operate at ambient conditions [8]. Graphene flakes of controllable properties can be formed without the need for volatile solvents or reducing agents. The process can take several minutes or hours to complete, and the reported results are encouraging for the fast processing of large quantities of graphene flakes [9]. The electrochemical method utilizes an electrolyte solution and electrical charges to drive structural expansion and exfoliation (beside oxidation or reduction), intercalation, and exfoliation on a piece of graphite (plate, rod, wire) to produce flakes. The experimental arrangement uses a monopolar, undivided electrolysis cell. The yield, productivity, and properties of graphene flakes can be finetuned by controlling electrolytes and electrolysis parameters [10,11,12]. Electro-exfoliation has some advantages over traditional chemical methods and is a promising method for the mass production of graphene [13]. To date, electrochemical exfoliation of graphite has been performed mainly in two different types of electrolytes, i.e., ionic liquids [12,14] and typical inorganic electrolytes, such as acids and their salts: HCl (e.g., NaCl, KCl), HF (e.g., NaF, KF), H_2_SO_4_ (e.g., Na_2_SO_4_, K_2_SO_4_, (NH_4_)_2_SO_4_,), HNO_3_ (e.g., NaNO_3_, KNO_3_, NH_4_NO_3_), H_3_PO_4_, NaClO_4_ [15,16,17,18,19,20,21,22,23,24]. Electro-exfoliation in non-electrolytes like H_2_O_2_ was likewise performed. The intercalation process, as well as gas evolution, may open up graphene sheets, causing their expansion and exfoliation of the graphene layers [25]. Exfoliation in acidic electrolytes yields graphene flakes of better quality and larger lateral size, but it is impossible to avoid a significant amount of oxygen-containing functional groups due to graphite being oxidized by the products of acid electrolysis [10,11,26]. On the other hand, exfoliation in ionic liquids results in a low yield of graphene with has a small lateral size and often functionalization with ionic liquids, disrupting the electronic properties of graphene [14,27]. Therefore, a proper electrolyte system should allow a proper balance between the high purity and high yield of these exfoliated graphene flakes.

The studies performed on graphene electro-exfoliation thus far have almost exclusively been focused on the graphene flake splitting itself and the chemical alterations of the flakes due to certain electrode reactions. The same studies overlooked the problem of graphene flakes’ secondary adhesion once the dispersing liquid phase (mainly water) was removed. This causes the electro-exfoliation to be a reversible phenomenon, making the outcome benefits (the presence of separate graphene flakes) hardly applicable in practice. No fixation method of split graphene flakes has been proposed so far.

In the recent study we have successfully elaborated a 3D structuring method for durable fixation of graphene flakes obtained from a commercial graphite by wet-chemistry exfoliation [28]. In situ precipitated Na_2_CO_3_ nanocrystals or CaCO_3_ nano-powder were used as a hard template. Graphene flakes were obtained by a wet chemistry exfoliation of commercial graphite. The flakes were premixed with a non-specific binder and the hard template and then carbonized at temperatures of 700 to 800 °C under the flow of nitrogen. The addition of a template allowed the surface area to increase up to 287 m^2^ g^−1^ for the Na_2_CO_3_ template and 333 m^2^ g^−1^ in the case of CaCO_3_, while the surface area of few m^2^ g^−1^ was noted for the raw graphite. The wet-chemistry method led to an efficient deglomeration of graphene flakes to double-layered (DLG) and few-layered (FLG) graphene.

Thus, durably fixed graphene flakes can form 3D graphene sponges. These sponges in turn can substitute typical porous carbon-based materials, e.g., activated carbons, in numerous applications exploiting enhanced specific surface area and porosity (gas/liquid phase adsorption, electrode processes, etc.). In this work, we demonstrate a highly effective electrochemical exfoliation method in aqueous inorganic salt sodium sulfonate (Na_2_SO_4_). We present the first case of graphene stabilized with a small amount of carbon obtained from poly(furfuryl alcohol). The graphene flakes which stabilize in a 3D structure were acquired by electroexfoliating two different graphite precursors: graphite foil and commercial graphite powder. Additionally, we assumed that split flakes can be durably separated from one another by nanoparticles (Na_2_CO_3_, CaCO_3_) which can be later removed to release nano-voids, i.e., pores. The influence of process parameters on the characteristics of produced graphene sponges was investigated.

## 2. Materials and Methods

### 2.1. Materials

Extra pure graphite was purchased from Merc. The graphene films/foils 240 μm thick were provided by Sigma Aldrich. Other reagents i.e., Na_2_SO_4_, CaCO_3_, furfuryl alcohol (FA), HCl and H_3_PO_4_ were of analytical purity grade and were obtained from Sigma Aldrich (Warsaw, branch in Poland), POCH (Gliwice, Poland) and SS Nano (Houston, TX, USA).

### 2.2. Electrochemical Exfoliation

Electrochemical exfoliation of extra-pure graphite (Merc) was performed first. Such an electro-exfoliation of a powder creates serious experimental obstacles, since the graphite powder is a loose material with no specific durable shape which could aid holding the mechanical electrode. Therefore, a special electrochemical experimental setup was developed in which the loose powder was electrically charged by temporary contact with a positively charged bottom electrode (platinum plate), as the particles were pulled downwards to the metal surface (sedimentation) by the force of gravity, as depicted in Figure 1a. The bottom electrode may be called a working electrode since it was in permanent contact with the graphite powder exposed to electro-exfoliation. A potential of +10 V was set on the electrode, which then played the role of an anode (expected oxidation of anions being present in the electrolyte). After pouring the electrolyte solution (water Na_2_SO_4_ 1 M solution), graphite powder was sunk and covered by a porous polymeric membrane/separator to avoid graphite/graphene powder spreading throughout the entire electrolyte volume. The electric circuit was completed by placing a movable polymeric plug containing the second platinum electrode (a mesh), on which ground potential was set (0 V) to enable its cathodic function. The movable plug/electrode provided an additional compressing force which, beside natural sedimentation, caused the graphite particles to be in electric contact with the anode, assuming the whole experimental setup is vertically oriented. The meshed structure of the upper electrode (cathode) was permeable for gases evolved during electro-exfoliation. The electro-exfoliation was performed for an experimentally-verified length of time. After this time passed, the electro-exfoliation was opened and the suspension (water, electrolyte residues, exfoliated graphene flakes, graphite residues) was poured into a beaker and subjected to other chemical manipulations. Additional force/weight may generally be placed on the movable plug, though it was not in the current study.

In the second step, we performed the electro-exfoliation of graphite foil (Sigma Aldrich) ca. 3.5 cm × ca. 3.5 cm in size. The foil, since it had a fixed shape, served as an anode once the potential of +10 V was set on it. Thus, the graphite itself played the role of anode while a platinum plate worked as the cathode (after having the ground potential set on it). Anode and cathode zones were not separated by a membrane in this case (Figure 1b). The same electrolyte was applied. Electric current passed through the system for an experimentally-verified period, after that both electrodes (platinum and graphite foil residues) were removed and the suspension (water, electrolyte residues, exfoliated graphene flakes, graphite residues) was poured into a beaker and subjected to other chemical manipulations. Spectacular exfoliation of the foil was observed after only a few minutes, i.e., loose black flakes split from the foil, forming a dark suspension while the foil electrode itself vanished gradually.

A direct current (DC) voltage (R&S^®^HMP2020 power supply) was applied between the platinum and graphite electrodes, and the electrolysis procedure lasted 2 h for graphite powder and 30 min for graphite foil at room temperature. During this process, substances expected to work as a separator of newly exfoliated graphene flakes (denoted later as templates) were added in one of two forms: a nano-powder of CaCO_3_ (5–40 nm diameter) or a saturated solution of Na_2_CO_3_.

Graphene flakes obtained through electrochemical graphite split were durably separated by the use of an additional agent, i.e., a template (nano-grains of Na_2_CO_3_ or CaCO_3_). Furfuryl alcohol was added along with H_3_PO_4_ (polycondensation catalyst) as a potential gluing agent. The resulting graphene–furfuryl alcohol (FA)-template mass was subjected to heat treatment to polymerize PA to poly(furfuryl alcohol (PFA) (1 h at 100 °C), then carbonized in a N_2_ atmosphere with a heating rate of 10 °C min^−1^ up to 600 °C. This temperature was maintained for 1 h. The process was carried out in a tubular furnace (Thermolyne F21100). After carbonization, the samples were treated with a concentrated (34%–37%) HCl solution for 20 min (12 mL of acid was used per 1 g of carbon) to remove CaCO_3_ and Na_2_CO_3_. They were then washed with distilled water on a Büchner funnel until the pH of the solution reached 6–7. It was dried again in an electric furnace at 100 °C for 24 h. Further in the text, EG_F_2 and EG_F_1 are used as the denotation of samples obtained from graphite foil with CaCO_3_ and Na_2_CO_3_, respectively. The EG_P_2 and EG_P_1 samples were obtained from graphite powder, using CaCO_3_ and Na_2_CO_3_, respectively.

### 2.3. Material Characterization

Structural parameters like the specific surface area (Brunauer–Emmett–Teller surface area, S_BET_) and pore structure of graphite samples were examined using the low-temperature nitrogen adsorption method. The relevant isotherms of all samples were measured at −196 °C using an automatic adsorption instrument, ASAP 2010 (Micromeritics, Norcross, GA, USA). Prior to gas adsorption measurements, the carbon materials were outgassed in a vacuum at 200 °C for 2 h. The pore size distribution was derived from adsorption branches by the nonlocalized density functional theory (NLDFT) method. The elemental composition (carbon, nitrogen, and hydrogen) of the materials was analyzed by means of a combustion elemental analyzer (Vario MACRO CHN, Elementar Analysensysteme GmbH). The morphology of the samples was analyzed using scanning electron microscopy (SEM, 1430 VP, LEO Electron Microscopy Ltd., Cambridge, UK) and high-resolution transmission electron microscopy (HRTEM, FEI Europe production, model Tecnai F20 X-Twin). The samples were examined using Raman spectroscopy–microscope: Renishaw InVia (Renishaw plc, Gloucestershire, UK), laser: Modu-Laser Stellar-REN, Multi-Line (maximum Power 150 mW), camera: Leica DM1300M Infinity 1, lens: Leica, N PLAN L50×/0.5. All spectra were collected at ambient temperatures with a 532 nm excitation wavelength. Atomic force microscopy (AFM) investigations were measured by Scanning Probe Microscope (SPM) produced by Veeco (Digital Instrument, Plainview, NY, USA).

## 3. Results and Discussion

Graphene flakes can be produced from graphite, either in the form of a foil or a powder electrode, by exfoliation in 1 M Na_2_SO_4_ solution, wherein the same electric charges placed on the graphite domains create repulsive forces. Additionally, when a positive potential is set on the working electrode (either graphite foil or graphite powder), anions (mainly SO_4_^2−^) absorb on active sites of the graphite electrodes and oxidize to gaseous oxygen. The anions and evolved gas may penetrate into the interstitial space between graphene layers (in powder or foil) and expand the materials to the point of splitting them into less agglomerated forms, like single-layered graphene (SLG) and few-layered graphene (FLG). Gas (mainly oxygen) evolution accelerates the switching rate of graphite to graphene.

We propose a mechanism of electrochemical exfoliation using an additional substrate of Na_3_CO_3_ (Figure 2a) or CaCO_3_ (Figure 2b). In the former procedure (Figure 2a), to a 1 M Na_2_SO_4_ electrolyte solution a small volume of N-methyl-2-pyrrolidone (NMP) and cetyltrimethylammonium chloride solution (CTAC) was added. After electrochemical exfoliation, the resulting product was ultrasonicated for 30 min, then a Na_2_CO_3_ solution at a concentration of about 16.9% was added; the mixture was left on a magnetic stirrer for 1 h, then sonicated again for 1h. The samples were then washed with distilled water on a Büchner funnel. To evaporate the distilled water, the mass was dried in an electric oven at 80 °C for 24 h. In the second method (Figure 2b), instead of Na_2_CO_3_, CaCO_3_ nanoparticles with a size range of 5–40 nm were added.

Raman spectroscopy is a method particularly useful for characterizing graphene/graphite type materials [29,30]. Among others, Raman spectroscopy makes it possible to distinguish graphite-based materials from graphene-based ones, to estimate graphene layers stacked together (SLG – Single Layered Graphene, FLG – Few Layered Graphene, and MLG – Multi Layered Graphene differentiation), and to resolve other problems. Figure 3a–d provides Raman spectra of graphene obtained in the two ways with two different separators, Na_2_CO_3_ and CaCO_3_. Additionally, Table 1 presents the position and intensity of the peaks in Figure 3. Figure 3b,d show the Raman spectra of the graphene products obtained with different separators, where the increased value of I_D_/I_G_ suggests the decrease of graphene agglomerate thickness. As shown in Table 1, a strong relative intensity of the 2D peak indicates fewer stacked layers, which in foil exfoliation with CaCO_3_ is beneficial for the formation of graphene. The presence of graphitic carbon in the EG_F_2 and EG_F_1 samples (Figure 3b) revealed the well-documented D and G bands at 1351 and 1579 cm^−1^, with an I_D_/I_G_ ratio of 0.53 and 0.13, respectively. An analysis of the Raman spectra for pristine graphite foil (Figure 3a) and pristine graphite powder (Figure 3c) leads us to believe that the foil is closer to an ideal graphene material than the powder. This conclusion results from the I_D_/I_G_ ratio, which is only 0.17 for the foil and 0.59 for the powder. Furthermore, the conclusion is supported by the observed shape of the D band, which is very broad in the powder spectrum.

To confirm the presence of double-to-few-layered stabilized graphene, SEM and HRTEM analysis was used. A SEM image of PFA-graphene is shown in Figure 4. SEM studies the fracture surface in pure graphene/carbon from PFA (Figure 4). Prior work has shown that graphene aggregates can be seen in SEM, but it is quite difficult to image individual dispersion graphene sheets in composites at low weight fractions. We observed no signs of aggregation. We could not observe individual graphene sheets in the fracture surface of graphene.

The HRTEM technique allowed the dispersion of graphene sheets and was stabilized with small amount of carbon from PFA. Thin-layer graphene sheets were produced at a high yield with large flake sizes (Figure 5). Few-layered graphene sheets can be seen in Figure 5. Images of the sheet edges are visible in Figure 5b and Figure 6d; those in Figure 5d indicate that the sheets are 2–4 or more layers thick. The HRTEM images allow to evaluate the size of electrochemically exfoliated graphene sheets as 0.7−4 µm.

In order to get information on the thickness and lateral size of electrochemically exfoliated graphene sheets, AFM analysis was performed using EG_P_2 and EG_F_2 samples subjected to sonication in ethanol. The results are presented in Figure 6. Typical graphene sheet thickness is about 2 nm. Assuming that the thickness of single layered graphene is 0.51 nm [31] and the interlayer distance is 0.34 nm the measured value corresponds to three layered graphene: 3 × 0.51 nm + 2 × 0.34 nm = 2.17 nm. This result is consistent with Raman spectroscopy studies which also suggest the performed electro-exfoliation delivers few-layered graphene mainly.

Generally, other communications on exfoliation provide no analysis of structural parameters such as specific surface area and pore structure. This problem may be regarded as a kind of blank spot which is yet to be filled. We assume that these structural parameters are difficult to measure when the exfoliated graphene flakes are not secured against secondary stacking (due to inter-flake attractive forces known as spontaneous pi–pi stacking). The current study is based on the separation (addition of nano-templates) and fixation (carbon bridges originating from PFA) of exfoliated flakes. Thus, in the case of these samples, i.e., graphene sponges, structural parameters like surface area and pore structure become measurable. Our study is unique among reports dealing with this problem.

The results regarding specific surface area (S_BET_) are given in Table 2 for the two different graphene-flake separators, i.e., Na_2_CO_3_ or CaCO_3_. The highest S_BET_ was found for samples obtained with CaCO_3_ as a separator. Both electro-exfoliation approaches (graphene foil and powder) yielded samples with the highest values of S_BET_, i.e., EG_F_2 with 121 m^2^ g^−1^ and EG_P_2 with 220 m^2^ g^−1^. These values were calculated using the regression of data from the original nitrogen adsorption at –196 °C. The same data let us estimate pore size distribution by using the widely accepted nonlocal-density functional theory (NLDFT) model [32,33,34]. Figure 3e,f show pore size distribution (PSD) of the EG_F_2, EG_F_1 and EG_P_2, EG_P_1 samples. Interestingly, all the PSD curves only cover the range of micropores. The PSD function for foil-originated samples is unique because of two overlapping peaks in which maxima occur at 9.3, 11.7 (Figure 3e). One may consider the distribution as almost monomodal, which is in contrast to the bimodal PSD function recorded for graphene sponges created by the electro-exfoliation of graphene powder. For two graphene samples obtained from graphite foil we can observe main peaks at 0.9, 1.2, and 1.7 nm (Figure 3e). PSD curves (Figure 3f) show that the dominant pore sizes were similar for both samples of graphene obtained from graphite powder at 0.8, 1.8 nm and 0.9, 1.7 nm of EG_P_2 and EG_P_1, respectively.

The chemical composition was investigated using CHN combustion elemental analysis and the results are presented in Table 2. Generally, the carbon content was very high and typical for real graphene-based materials [22], i.e., it is proven that total carbon content ranges from 90.0wt% to 94.3wt%. Combustion elemental analysis also delivers information on the content of oxygen. Assuming that the content of unidentified elements (complement to 100% for each sample) is the content of oxygen, the carbon-to-oxygen ratio (C/O ratio) examined by elemental analysis is high (from 11.2 to 17.8) and typical to graphene not to graphene oxide [15,22]. Therefore, the obtained sponges can be considered to consist of graphene flakes. The C/O ratio is considerably higher for the foil-originated samples, which is in accordance with the Raman spectroscopy results, proving that the foil likely consists of graphene domains. Some XPS (X-Ray Photoemission Spectroscopy) studies as well as SEM-EDX studies (Scanning Electron Microscopy combined with Energy Dispersive X-Ray module) did not reveal the content of heavier elements (like Ca, Na, P, etc.) is higher than 1%. Thus, our assumption that unidentified element content (Table 2) may be ascribed to oxygen is justified and realistic.

## 4. Conclusions

In conclusion, an accessible method was developed and demonstrated for obtaining few-layered graphene directly from graphite by electro-exfoliation and 3D stabilization by means of carbonized PFA as a stabilizer and certain inorganic carbonates as templates. The obtained samples can be treated as porous carbon sponges built of graphene flakes. The study indicates that the presence of carbonized PFA is critical for stabilizing the graphene and improving the surface parameters; the specific surface area increased several times compared to the starting material (graphite). The relatively high graphene purities and excellent stabilization with PFA show both methods to be promising ways of developing porous graphene. The morphological properties include the nanostructure of graphene powder with different sizes, characterized by SEM and HRTEM. The structural properties indicate a high quality for graphene, as determined by Raman spectroscopy. Low-cost and environmentally friendly production of such high-quality graphene is important, not only for future-generation electronics, but also for large-scale applications, such as composite materials, supercapacitors, fuel-cells, and batteries. Any practical application requires upscaling of the electrochemical arrangement and a cost calculation. The economic aspect of electrochemical exfoliation directly points to powdered graphite of high purity since its market prices are definitely much lower than the price of graphene foil. Experimental setup for exfoliation of graphite powder which was tested in the current project can be enlarged as single cell, but also small volume single cells may be used as a cell array. The presented electro-exfoliation approach has one general advantage over typical wet chemistry approaches: the inevitable chemical reagents i.e., electrolytes are common, inexpensive, and relatively easy to utilize. An upscale of the electrochemical cell will be the target of subsequent studies.

## Figures and Tables

**Figure 1 nanomaterials-09-00971-f001:**
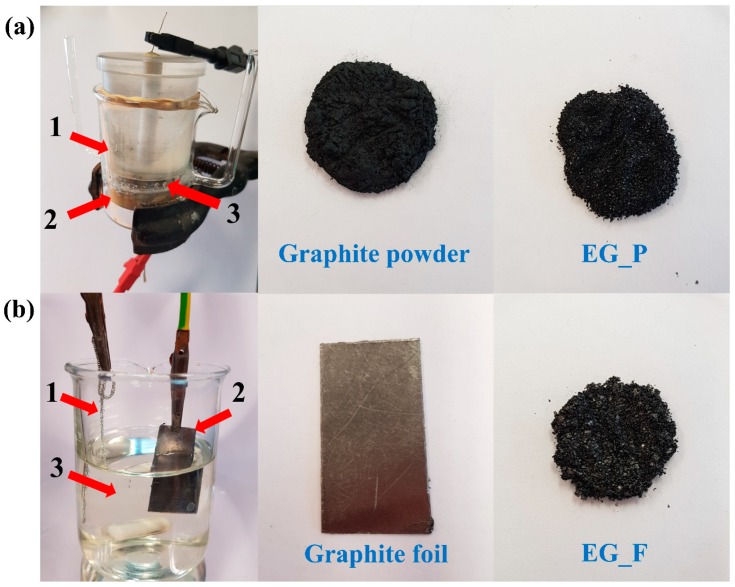
Experimental setup of electrochemical exfoliation technique for: (**a**) graphite powder, where: (1) platinum electrode (cathode), (2) platinum electrode (anode), (3) space for the electrolyte and graphite for exfoliation, image of initial graphite powder and the final graphene after electrochemical exfoliation (EG_P); (**b**) graphite foil, where: (1) platinum electrode (anode), (2) graphite foil (cathode), (3) electrolyte, image of initial graphite foil and the final graphene after electrochemical exfoliation (EG_F).

**Figure 2 nanomaterials-09-00971-f002:**
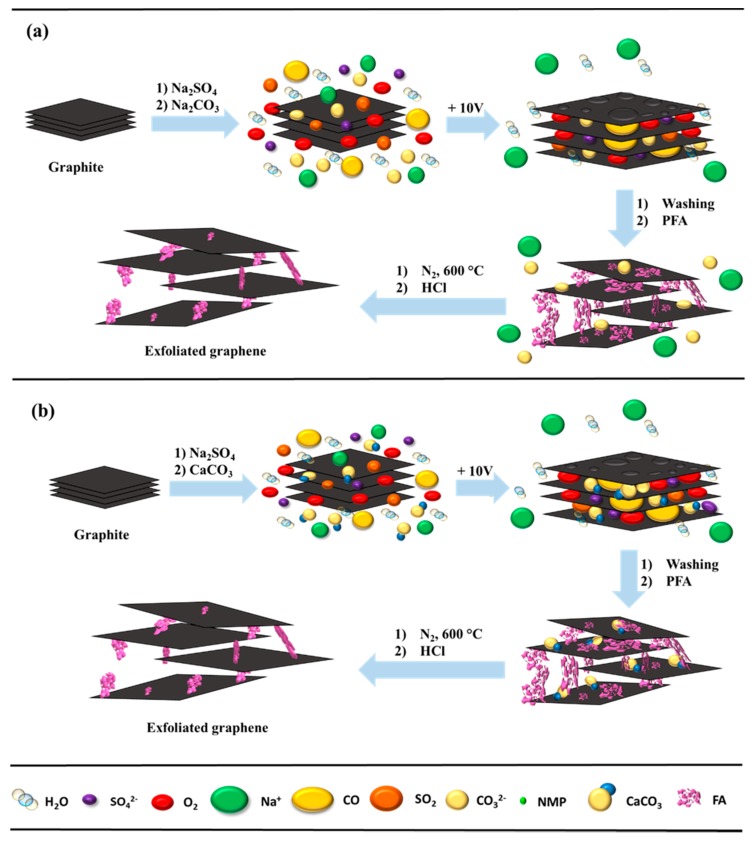
Schematic illustration of the mechanism of electrochemical cathodic exfoliation with: (**a**) Na_2_CO_3_ and (**b**) CaCO_3_ as a separator and poly(furfuryl alcohol) (PFA) as a stabilizer.

**Figure 3 nanomaterials-09-00971-f003:**
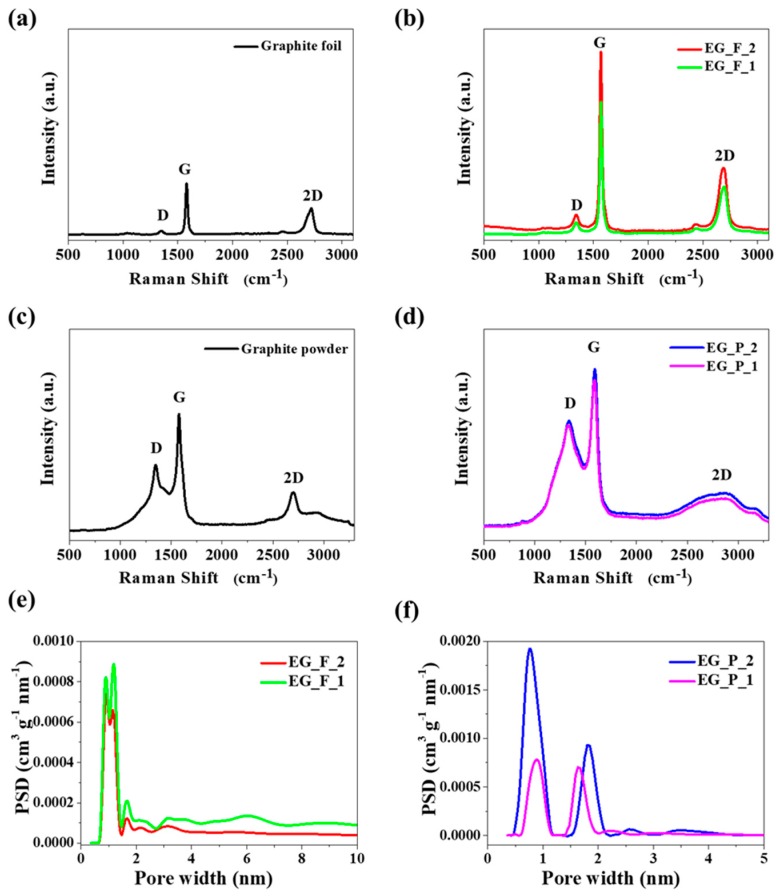
Raman spectra of (**a**) graphite foil, (**b**) EG_F_2 and EG_F_1, (**c**) graphite powder, (**d**) EG_P_2 and EG_P_1 samples. Pore size distribution curves of graphene samples obtained from graphite (**e**) foil and (**f**) powder.

**Figure 4 nanomaterials-09-00971-f004:**
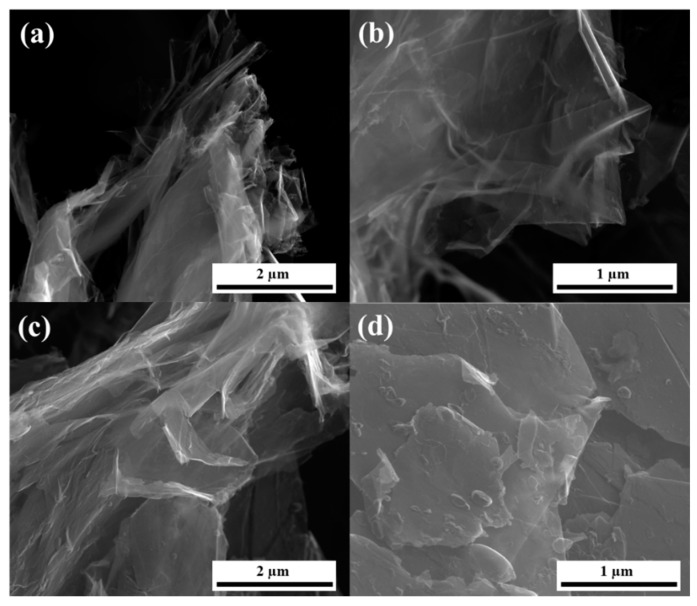
SEM images of (**a**,**b**) EG_F_2 and (**c**,**d**) EG_F_1.

**Figure 5 nanomaterials-09-00971-f005:**
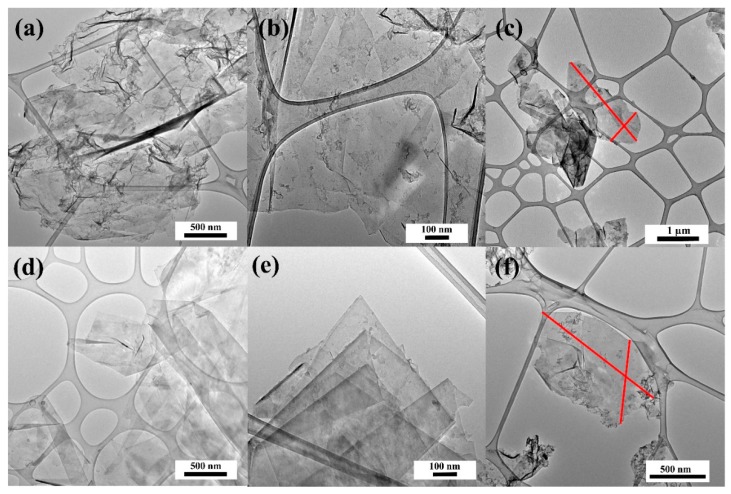
High-resolution transmission electron microscopy (HRTEM) images of (**a**–**c**) EG_F_2 and (**d**–**f**) EG_F_1.

**Figure 6 nanomaterials-09-00971-f006:**
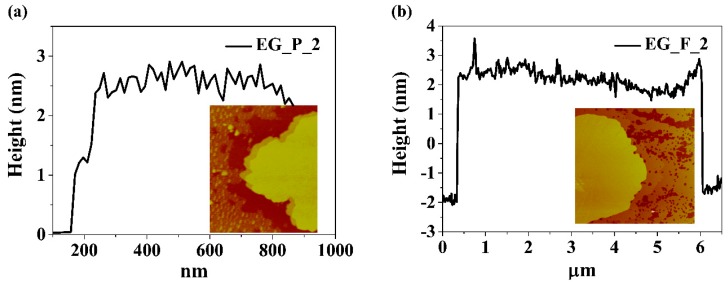
Atomic force microscopy (AFM) analysis together with height profile of (**a**) EG_P_2, (**b**) EG_F_2.

**Table 1 nanomaterials-09-00971-t001:** Raman peak positions and intensities for the products obtained from samples in different series.

Sample	D peak		G peak		2D peak		I_D_/I_G_	I_2D_/I_G_
	(cm^−1^)	I	(cm^−1^)	I	(cm^−1^)	I		
Graphite foil	1349.3	694	1582.3	4158	2718.0	2343	0.17	0.56
Graphite powder	1343.4	1781	1578.0	2994	2705.6	1138	0.59	0.38
EG_P_2	1334.7	2868	1592.2	4081	2867.5	1207	0.70	0.30
EG_P_1	1330.1	2764	1587.6	3822	2868.7	1066	0.72	0.28
EG_F_2	1339.1	6058	1578.0	11,534	2705.6	4211	0.53	0.37
EG_F_1	1346.2	1343	1568.9	10,186	2692.1	3888	0.13	0.38

**Table 2 nanomaterials-09-00971-t002:** Specific surface area (S_BET_) and elemental content of carbon, hydrogen, and nitrogen.

Sample	S_BET_ (m^2^ g^−1^)	Content (wt%)		
		C %	H %	N %
EG_F_2	121	93.0	0.6	0.2
EG_F_1	52	94.3	0.3	0.1
EG_P_2	220	90.0	1.8	0.2
EG_P_1	103	90.5	2.0	0.2

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
