# Peer review of "Electro-Exfoliation of Graphite to Graphene in an Aqueous Solution of Inorganic Salt and the Stabilization of Its Sponge Structure with Poly(Furfuryl Alcohol)"

_nanomaterials, 2019, doi:10.3390/nano9070971_

Reviewer 1 Report

The manuscript describes a method to prepare graphene from electroexfoliation of graphite. Starting from graphite materials, the authors utilized electrochemical reaction in various electrolytes to exfoliate graphite for graphene sponge generation. The resulting graphene sponges were then characterized with various analytical tools, such as Raman spectroscopy, SEM, HRTEM, elemental analysis, and adsorption to determine the chemical composition, microstructure, and morphology. The experimental results showed that high quality graphene were generated with this method. The authors presented the experimental data with good scientific insights and figure presentation. Thus, it is recommended for publication with minor revision. Some suggestions and comments are listed below:

1. the authors should give an optical picture of the initial graphite material and the final graphene sponge in either figure 1 or 2. It will be better for readers to understand the process.

2. Can the authors also provide AFM results to evaluate the thickness of the graphene thickness? Also please roughly evaluate the size of graphene sheets in the TEM or SEM images and discuss the differences between samples.

3. please compare the surface area provided in this method with others in the literature.

4. In table 2, it seems that there are some elements other than C H N in the graphene samples, and the percentage is ~5-8%. Although the authors explain it might be oxygen, but conflict with the observed low oxide results. The authors’ explanation in line 256-261 is not very convincing. Please rephrase to clarify the results or make another elemental analysis for Ca or Na since there could be some residual materials attached to the sponge.

Author Response

On behalf of all authors I would like to thank for a careful reading of our manuscript and constructive remarks. We intend to improve our manuscript in accordance with these comments.  Our responses to particular remarks are given below and are marked with red color. Similarly, we used red color to see the revised parts of our manuscript.

Sincerely your

Jerzy P. Lukaszewicz, corresponding author

RESPONSES:

Reviewer 1

1. The authors should give an optical picture of the initial graphite material and the final graphene sponge in either figure 1 or 2. It will be better for readers to understand the process.

Response: We thank the referee for this suggestion. On Fig. 1 we added additional images of:

- initial graphite powder and the final graphene after electrochemical exfoliation (EG_P),

- image of initial graphite foil and the final graphene after electrochemical exfoliation (EG_F), (Page 4).

2. Can the authors also provide AFM results to evaluate the thickness of the graphene thickness? Also please roughly evaluate the size of graphene sheets in the TEM or SEM images and discuss the differences between samples.

Response: We thank the referee for this suggestion. We added description of AFM instrument (page 5) and results of AFM analysis . We add additional images on Fig. 6 and description about size of graphene sheets (Page 9).

We add two additional HRTEM images to Fig. 5, to better describe size of graphene sheets (Pages 8 and 9).

3. Please compare the surface area provided in this method with others in the literature.

Response:

The problem of specific surface area of exfoliated graphene is consequently omitted in known to us papers describing electroexfoliation of graphen. We did not find such values and we can’t cite them.

4. In table 2, it seems that there are some elements other than C H N in the graphene samples, and the percentage is ~5-8%. Although the authors explain it might be oxygen, but conflict with the observed low oxide results. The authors’ explanation in line 256-261 is not very convincing. Please rephrase to clarify the results or make another elemental analysis for Ca or Na since there could be some residual materials attached to the sponge.

Response: We thank the referee for this suggestion. We have performed SEM-EDX studies which did not revealed a considerable presence of heavy elements is the materials under investigation. According to these studies the materials consist of C, N, O (H is not visible in EDX studies) and less than 1% of other elements (Ca, Na etc.). However, we have modified the text to make it more convincing.

Reviewer 2 Report

The manuscript is rather well-written.

One possible way to improve it might be to include some information on what applications might benefit form this technique, and/or wether this is feasible for large scale production of graphene sponges.

Author Response

On behalf of all authors I would like to thank for a careful reading of our manuscript and constructive remarks. We intend to improve our manuscript in accordance with these comments.  Our responses to particular remarks are given below and are marked with red color. Similarly, we used red color to see the revised parts of our manuscript.

Sincerely your

Jerzy P. Lukaszewicz, corresponding author

Reviewer 2

The manuscript is rather well-written.

One possible way to improve it might be to include some information on what applications might benefit from this technique, and/or wether this is feasible for large scale production of graphene sponges.

Response:

A comment is added in „conclusions”.

Reviewer 3 Report

Achieving exfoliated graphene is a hot topic and this manuscript provides guidance for further advances in materials science, especially for the fabrication of nanostructured materials. The paper is well-written and fairly organized. I recommend it for publication after minor revision.

In the introduction, an overview on the methods for exfoliating graphite and, generally speaking, graphene-based nanocarbons should be added. This deature would improve the readers' interest.

Some papers dealing with liquid exfoliation of graphite and/or with graphene dispersion should be added in the reference list: 

1) Composites Part B: Engineering 168, pp. 550-559 (2019)

2) Composites Science and Technology 156, pp. 166-176 (2018)

3) Polymers 11(5),857 (2019)

4) Chemical Engineering Journal 308, pp. 1034-1047 (2017)

In the Experimental section, a subsection dedicated on raw materials and chemical reactants would be preferable.

Furthermore, if the authors have the possibility to carry out other types of spectroscopic analysis, adding at last one more characterization technique among FTIR, XPS, UV-vis spectroscopy could corroborate the interesting results of Raman and elemental analyses.

Author Response

On behalf of all authors I would like to thank for a careful reading of our manuscript and constructive remarks. We intend to improve our manuscript in accordance with these comments.  Our responses to particular remarks are given below and are marked with red color. Similarly, we used red color to see the revised parts of our manuscript.

Sincerely your

Jerzy P. Lukaszewicz, corresponding author

Reviewer 3

Some papers dealing with liquid exfoliation of graphite and/or with graphene dispersion should be added in the reference list: 

1) Composites Part B: Engineering 168, pp. 550-559 (2019)

2) Composites Science and Technology 156, pp. 166-176 (2018)

3) Polymers 11(5),857 (2019)

4) Chemical Engineering Journal 308, pp. 1034-1047 (2017)

Response: We thank the referee for this suggestion. We added these references to our manuscript (Page 1).

In the Experimental section, a subsection dedicated on raw materials and chemical reactants would be preferable.

Response:

We have added a brief information in subsection 2.1.

Reviewer 4 Report

The manuscript provides a systematic study of the preparing exfoliate graphite through electrochemical method. The authors have projected two different methodologies and have shown that they can avoid restacking of graphene layers by the incorporation of a polymeric substance. Though the projected results are interesting and will arise interest in the audience, there are a few concerns.

The authors should detail why the have chosen this particular polymer out of a variety of polymers.

The authors haven't optimized the voltages,  concentration and temperature of the bath they have used in this study. If they are using data from other resources, it has to be mentioned in the reference. 

The quantity of material taken and the quantity of exfoliated graphite obtained has to be mentioned, with out which the ease of this technique cannot be understood.

The authors have to repeat the experiment in duplicate or triplicate to understand the viability of the method. A mere single time experiment and the results there off will be highly localized.

The experimental section has to be made more clear. Its not understandable whether the PFA treatment is carried out only for the graphite electrode electrochemical exfoliation or also for the powder exfoliation. 

The order of presenting the results should be kept uniform through out the manuscript.

Author Response

On behalf of all authors I would like to thank for a careful reading of our manuscript and constructive remarks. We intend to improve our manuscript in accordance with these comments.  Our responses to particular remarks are given below and are marked with red color. Similarly, we used red color to see the revised parts of our manuscript.

Sincerely your

Jerzy P. Lukaszewicz, corresponding author

RESPONSES:

Reviewer 1

1. The authors should give an optical picture of the initial graphite material and the final graphene sponge in either figure 1 or 2. It will be better for readers to understand the process.

Response: We thank the referee for this suggestion. On Fig. 1 we added additional images of:

- initial graphite powder and the final graphene after electrochemical exfoliation (EG_P),

- image of initial graphite foil and the final graphene after electrochemical exfoliation (EG_F), (Page 4).

2. Can the authors also provide AFM results to evaluate the thickness of the graphene thickness? Also please roughly evaluate the size of graphene sheets in the TEM or SEM images and discuss the differences between samples.

Response: We thank the referee for this suggestion. We added description of AFM instrument (page 5) and results of AFM analysis . We add additional images on Fig. 6 and description about size of graphene sheets (Page 9).

We add two additional HRTEM images to Fig. 5, to better describe size of graphene sheets (Pages 8 and 9).

3. Please compare the surface area provided in this method with others in the literature.

Response:

The problem of specific surface area of exfoliated graphene is consequently omitted in known to us papers describing electroexfoliation of graphen. We did not find such values and we can’t cite them.

4. In table 2, it seems that there are some elements other than C H N in the graphene samples, and the percentage is ~5-8%. Although the authors explain it might be oxygen, but conflict with the observed low oxide results. The authors’ explanation in line 256-261 is not very convincing. Please rephrase to clarify the results or make another elemental analysis for Ca or Na since there could be some residual materials attached to the sponge.

Response: We thank the referee for this suggestion. We have performed SEM-EDX studies which did not revealed a considerable presence of heavy elements is the materials under investigation. According to these studies the materials consist of C, N, O (H is not visible in EDX studies) and less than 1% of other elements (Ca, Na etc.). However, we have modified the text to make it more convincing.        

Reviewer 2

The manuscript is rather well-written.

One possible way to improve it might be to include some information on what applications might benefit from this technique, and/or wether this is feasible for large scale production of graphene sponges.

Response:

A comment is added in „conclusions”.

Reviewer 3

Some papers dealing with liquid exfoliation of graphite and/or with graphene dispersion should be added in the reference list: 

1) Composites Part B: Engineering 168, pp. 550-559 (2019)

2) Composites Science and Technology 156, pp. 166-176 (2018)

3) Polymers 11(5),857 (2019)

4) Chemical Engineering Journal 308, pp. 1034-1047 (2017)

Response: We thank the referee for this suggestion. We added these references to our manuscript (Page 1).

In the Experimental section, a subsection dedicated on raw materials and chemical reactants would be preferable.

Response:

We have added a brief information in subsection 2.1.

Furthermore, if the authors have the possibility to carry out other types of spectroscopic analysis, adding at last one more characterization technique among FTIR, XPS, UV-vis spectroscopy could corroborate the interesting results of Raman and elemental analyses. 

Response:

Generally, there is a problem with spectroscopic methods suitable and informative for graphene based materials characterization. Some of them are in fact apparently useful. We have already added AFM results to the revised version of our manuscript. In fact, at the preliminary stage of our work with graphene, we did some FT-IR studies concerning similar materials. To our experience with FT-IR the results are informative but very little only since non-oxidized graphene (our materials) do not contain much of heteroatoms and specific functional groups like –OH, -C-O-C-, -CHO, -COOH etc. One may only state that such species may be present in residual quantities. XPS is a similar case. Some information on XPS and FT-IR studies we have presented at few conferences as a side research material only. UV-VIS in case of graphene based materials is useful for the determination of forbidden energy gap width. That might be useful for characterization of electric/electronic properties of a material but it is not the main aim of current studies which are focused on exfoliation process itself and 3D structuring of exfoliated graphene flakes as well. Thus, Raman spectroscopy with its limitations remains a key characterization method and we applied it in this study.